

# Nectar-living yeasts of a tropical host plant community: diversity and effects on community-wide floral nectar traits

Azucena Canto[1], Carlos M. Herrera[2,*] and Rosalina Rodriguez[1,*]

[1] Centro de Investigacion Cientifica de Yucatan, A.C., Merida, Yucatan, Mexico
[2] Estación Biológica de Doñana, CSIC, Sevilla, Spain
[*] These authors contributed equally to this work.

## ABSTRACT

We characterize the diversity of nectar-living yeasts of a tropical host plant community at different hierarchical sampling levels, measure the associations between yeasts and nectariferous plants, and measure the effect of yeasts on nectar traits. Using a series of hierarchically nested sampling units, we extracted nectar from an assemblage of host plants that were representative of the diversity of life forms, flower shapes, and pollinator types in the tropical area of Yucatan, Mexico. Yeasts were isolated from single nectar samples; their DNA was identified, the yeast cell density was estimated, and the sugar composition and concentration of nectar were quantified using HPLC. In contrast to previous studies from temperate regions, the diversity of nectar-living yeasts in the plant community was characterized by a relatively high number of equally common species with low dominance. Analyses predict highly diverse nectar yeast communities in a relatively narrow range of tropical vegetation, suggesting that the diversity of yeasts will increase as the number of sampling units increases at the level of the species, genera, and botanical families of the hosts. Significant associations between specific yeast species and host plants were also detected; the interaction between yeasts and host plants impacted the effect of yeast cell density on nectar sugars. This study provides an overall picture of the diversity of nectar-living yeasts in tropical host plants and suggests that the key factor that affects the community-wide patterns of nectar traits is not nectar chemistry, but rather the type of yeasts interacting with host plants.

Corresponding author
Azucena Canto, azucanto@cicy.mx

## INTRODUCTION

Floral nectars are sugar-rich environments that frequently harbor distinctive microbial communities. Studies on microbial diversity conducted by *Brysch-Herzberg (2004)*, *Pozo, Herrera & Bazaga (2011)*, *Álvarez Pérez & Herrera (2013)*, *Jacquemyn et al. (2013)* and *Mittelbach et al. (2015)* revealed that floral nectar is frequently colonized by specialized sugar-consuming yeasts in the Ascomycota and Basidiomycota phyla, along with several bacterial groups. However, most studies of nectar-living microorganisms have been conducted in temperate areas; knowledge of nectar microbial diversity in tropical habitats remains poor. Only three preliminary assessments of the frequency of microbial cells in

floral nectars in several tropical environments have been conducted to date (*Herrera et al., 2009*; *Canto & Herrera, 2012*; *Belisle et al., 2014*). Altogether, these studies showed that the incidence of microorganisms in tropical nectars was higher than in temperate areas, and provided a glimpse of the high diversity harbored in tropical host plant communities. Diversity assessments in tropical nectars are still necessary to obtain a more complete view of the microbial distribution linked to nectars across different environments and latitudes. Another aspect of the impact of nectar-microbial diversity is that microorganisms can account for a significant fraction of community-wide variance in nectar traits, since the presence of yeast cells alters nectar sugar composition and concentration (the microbial imprint; *Canto & Herrera, 2012*). Evidence indicates that differential yeast effects on nectars are associated with characteristics of plants (type of nectar) and pollinator types. For example, pollinators are the main source of inocula for the initial establishment of microbial communities in nectars as they introduce their mouthparts into the nectaries in search of nectar rewards (*Canto et al., 2008*). The initial assemblage of microorganisms colonizing a flower will therefore depend largely upon the type of pollinator visiting host plants (*Belisle, Peay & Fukami, 2012*; *de Vega & Herrera, 2013*; *Mittelbach et al., 2015*). However, after initial colonization, the order of yeast species arrival to nectar and other nectar features strongly influence the growth of subsequent microorganisms, allowing some species to thrive but not others. The consequence is that the resulting microbial community consists of a cluster of phylogenetically related species (*Herrera et al., 2010*; *Belisle, Peay & Fukami, 2012*; *Vannette & Fukami, 2014*). In each community of nectariferous plants, nonrandom plant-microorganism associations can produce a mosaic of different qualities of floral nectars at the community level, with potential effects on plant–pollinator interactions (*Canto & Herrera, 2012*).

To characterize the diversity of nectar-living microorganisms in a tropical environment and to gain insights on factors driving community-wide variance in nectar traits, we analyzed the assemblage of yeast and yeast-like species (hereafter collectively termed 'yeasts') in floral nectars of tropical environments of the Yucatan peninsula, Mexico. By isolating and identifying culturable yeasts from the floral nectar of many animal-pollinated plants species and individuals, quantifying their population densities, and estimating nectar sugar concentration and composition, we will specifically assess (1) how diverse the community of nectar-living yeasts is in a tropical host plant community and between hierarchical sampling levels, (2) the existence of predictable associations between nectar yeasts and host plants, and (3) the differential impact of yeasts on nectar sugar composition associated with different host plants. Yeast diversity is discussed in relation to the different nectars sampled and the role of host plant types and types of yeasts in associations between plants and yeasts, all of which ultimately influence plant–pollinator interactions. Our results predict the existence of a relatively highly diverse assemblage of nectar-living yeasts, showing significant correspondence with the diversity of their host plants, as well as a significant impact of the interaction between yeasts and host plants in the effects that yeasts exert on floral nectars.

## MATERIALS AND METHODS

### Study area

Field sampling was conducted from September 2008 to November 2009 at 28 localities in an area of tropical vegetation (approx. 430 km$^2$) located between Chuburna and Dzilam de Bravo towns and the Cuxtal Ecological Reserve in north-western Yucatan, Mexico. The study area includes coastal dunes and adjacent dry forest environments, with elevation ranging between 1 m and 10 m. The climate is semi-arid in the coastal dune strip and subtropical in the dry forest, with a mean temperature of 26 °C in both areas and annual rainfalls of 370 mm and 1,077 mm, respectively. The vegetation is a low, open scrub dominated by xerophytes, halophyte herbs, thorny bushes, palms and 1–3 m treelets growing on sandy, nutrient-poor soils in the dune strip. The dry forest is made up of cacti, thorny shrubs and deciduous medium-height trees (3–8 m tall) growing on limestone bedrock soil with a thick litter layer (*Chan-Vermont, Rico-Gray & Flores, 2002*; *Canto & Herrera, 2012*). Permission to collect from natural areas of the Yucatan was granted by Secretaría del Medio Ambiente y Recursos Naturales, Delegación Yucatán-Subsecretaría de Gestion para la Protección Ambiental: Dirección General de Vida Silvestre (oficio 00837/09).

### Sampling method

To provide an overall picture of the diversity of nectar-living yeasts in floral nectars of the area, nectar samples were obtained from 18 host plant species belonging to 14 genera and 10 botanical families (Table 1), representing the diversity of life forms, flower shapes, pollinator types and taxonomic categories in the area. Plant species were individually sampled at their respective flowering peak, including as many flowering periods throughout the year as possible. At each locality, a single plant species was sampled (typically only one plant species was flowering in each place at the time of nectar collection), with the exception of coastal dune environments where nectar collection was performed at several sites and times. We adopted a five-tiered series of hierarchically nested sampling units for nectar collection, namely nectar samples or drops (Drop), individual plants (Individual), plant species (Species), plant genus (Genus), and botanical family (Family). Individual plants for nectar collection were chosen at random from the individuals growing at the locality. The criteria for collecting nectar samples from each individual plant was that flowers were approximately the same age, already open at the time of collection, but not wilted. This allowed for flowers to be exposed to prior pollinator visitation and the nectar to have been colonized by yeasts. Three single nectar samples (drops) were extracted from each flower using sterile microcapillary tubes with a calibrated scale of volume (Drummond Scientific, Broomall, PA, USA). The volume of nectar drops ranged from <0.50 to 1 µL. Flowers used in the sampling were fully open at the time of nectar collection. Three to six flowers were sampled from each plant and 6–10 individual plants were surveyed per plant species. Of the three nectar drops obtained from each flower, one was used for DNA-based identification of yeasts, another for quantification of yeast cell density and the other to estimate sugar composition and concentration, using methods described below (see Appendix S1 for further details on the numbers of nectar drops used in each method).

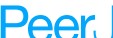

**Table 1 Nectar-living yeasts obtained from floral nectar of tropical plants.** Yeast species isolated from floral nectar in a tropical community of host-plants. Plant species, their respective botanical family, as well number of nectar samples or drops collected by plant species and individuals are reported along with frequency of each yeast species in nectar samples and accession numbers.

| Plant species (Botanical family) | Designation | *n* | Accession numbers | |
|---|---|---|---|---|
| | | | **CICY** | **NCBI** |
| *Agave angustifolia* (Agavaceae) | *Candida sorbosivorans* | 1 | CICYRN019 | KX908033 |
| | *Clavispora lusitaniae* | 4 | CICYRN016 | KX908034 |
| | *Hannaella siamensis* | 3 | CICYRN007 | KX908035 |
| *Bravaisia berlandieriana* (Acanthaceae) | *Papilotrema flavescens* | 3 | CICYRN004 | KX908036 |
| | *Cryptococcus* sp.1 | 1 | CICYRN011 | KX908037 |
| *Cordia sebestena* (Boraginaceae) | *Candida apicola* | 1 | CICYRN065 | KX908038 |
| *Gossypium hirsutum* (Malvaceae) | *Candida versatilis* | 1 | CICYRN061 | KX908039 |
| | *Vishniacozyma taibaiensis* | 1 | CICYRN053 | KX908040 |
| | *Starmerella bombicola* | 1 | CICYRN055 | KX908041 |
| | *Sympodiomycopsis paphiopedili* | 1 | CICYRN063 | KX908042 |
| *Gymnopodium floribundum* (Polygonaceae) | *Candida sorbosivorans* | 1 | CICYRN041 | KX908043 |
| | *Cryptococcus laurentii* var. *laurentii* | 1 | CICYRN040 | KX908044 |
| | *Papilotrema flavescens* | 1 | CICYRN039 | KX908045 |
| *Ipomoea crinicalyx* (Convolvulaceae) | *Candida etchellsii* | 1 | CICYRN313 | KX908046 |
| | *Candida powellii* | 1 | CICYRN303 | KX908047 |
| | *Metschnikowia ipomoeae* | 2 | CICYRN320 | KX908048 |
| | *Metschnikowia lochheadii* | 8 | CICYRN304 | KX908049 |
| | *Metschnikowia* sp. | 1 | CICYRN310 | KX908050 |
| | *Starmerella* sp. | 2 | CICYRN337 | KX908051 |
| | *Wickerhamiella occidentalis* | 1 | CICYRN341 | KX908052 |
| *Ipomoea hederifolia* (Convolvulaceae) | *Cryptococcus laurentii* var. *laurentii* | 3 | CICYRN225 | KX908053 |
| | *Hannaella sinensis* | 1 | CICYRN264 | KX908054 |
| | *Pseudozyma* sp. | 1 | CICYRN249 | KX908055 |
| | *Sympodiomycopsis paphiopedili* | 2 | CICYRN325 | KX908056 |
| | *Ustilago* sp. | 3 | CICYRN228 | KX908057 |
| | *Ustilago sparsa* | 6 | CICYRN256 | KX908058 |
| *Ipomoea nil* (Convolvulaceae) | *Cryptococcus* sp.1 | 1 | CICYRN217 | KX908059 |
| | *Saitozyma flava* | 1 | CICYRN207 | KX908060 |
| | *Sporidiobolus ruineniae* | 1 | CICYRN201 | KX908061 |
| | *Sympodiomycopsis paphiopedili* | 1 | CICYRN218 | KX908062 |
| | *Ustilago* sp. | 1 | CICYRN180 | KX908063 |
| | *Wickerhamiella occidentalis* | 1 | CICYRN182 | KX908064 |
| *Ipomoea triloba* (Convolvulaceae) | *Saitozyma flava* | 1 | CICYRN280 | KX908065 |
| | *Ustilago* sp. | 3 | CICYRN277 | KX908066 |
| | *Pseudozyma* sp. | 1 | CICYRN286 | KX908067 |
| *Lonchocarpus longistylus* (Fabaceae) | *Metschnikowia* sp. | 3 | CICYRN002 | KX908068 |
| *Malvaviscus arboreus* (Malvaceae) | *Candida versatilis* | 1 | CICYRN058 | KX908069 |

| Plant species (Botanical family) | Designation | n | Accession numbers | |
|---|---|---|---|---|
| | | | **CICY** | **NCBI** |
| *Merremia aegyptia* (Convolvulaceae) | *Aureobasidium* sp. | 1 | CICYRN221 | KX908070 |
| | *Papilotrema nemorosus* | 1 | CICYRN208 | KX908071 |
| | *Priceomyces melissophilus* | 1 | CICYRN210 | KX908072 |
| | *Sympodiomycopsis paphiopedili* | 1 | CICYRN209 | KX908073 |
| *Merremia dissecta* (Convolvulaceae) | *Cryptococcus laurentii* var. *laurentii* | 4 | CICYRN105 | KX908074 |
| | *Cryptococcus* sp.2 | 6 | CICYRN166 | KX908075 |
| | *Cryptococcus* sp.3 | 1 | CICYRN179 | KX908076 |
| | *Hannaella siamensis* | 3 | CICYRN107 | KX908077 |
| | *Papilotrema rajasthanensis* | 1 | CICYRN169 | KX908078 |
| | *Rhodotorula paludigena* | 1 | CICYRN188 | KX908079 |
| | *Sporidiobolus ruineniae* | 4 | CICYRN109 | KX908080 |
| | *Ustilago* sp. | 1 | CICYRN177 | KX908081 |
| *Operculina pinnatifida* (Convolvulaceae) | *Candida parazyma* | 1 | CICYRN165 | KX908082 |
| | *Cryptococcus laurentii* var. *laurentii* | 4 | CICYRN132 | KX908083 |
| | *Hannaella siamensis* | 1 | CICYRN134 | KX908084 |
| | *Kwoniella mangrovensis* | 2 | CICYRN127 | KX908085 |
| | *Metschnikowia ipomoeae* | 4 | CICYRN161 | KX908086 |
| | *Metschnikowia lachancei* | 1 | CICYRN155 | KX908087 |
| | *Metschnikowia lochheadii* | 3 | CICYRN144 | KX908088 |
| | *Metschnikowia* sp. | 2 | CICYRN150 | KX908089 |
| | *Rhodotorula paludigena* | 1 | CICYRN185 | KX908090 |
| | *Wickerhamiella occidentalis* | 1 | CICYRN137 | KX908091 |
| *Opuntia dillenii* (Cactaceae) | *Kurtzmaniella cleridarum* | 12 | CICYRN094 | KX908092 |
| | *Candida etchellsii* | 1 | CICYRN080 | KX908093 |
| *Passiflora foetida* (Passifloraceae) | *Candida bombi* | 1 | CICYRN051 | KX908094 |
| | *Candida sorbosivorans* | 3 | CICYRN014 | KX908095 |
| *Piscidia piscipula* (Fabaceae) | *Vishniacozyma taibaiensis* | 3 | CICYRN042 | KX908096 |
| | *Naganishia liquefaciens* | 6 | CICYRN046 | KX908097 |
| | *Sympodiomycopsis paphiopedili* | 3 | CICYRN048 | KX908098 |
| *Tecoma stans* (Bignoniaceae) | *Metschnikowia koreensis* | 13 | CICYRN036 | KX908099 |
| | *Metschnikowia ipomoeae* | 3 | CICYRN027 | KX908100 |
| | *Cryptococcus* sp.2 | 1 | CICYRN024 | KX908101 |

## Yeast isolation and DNA identification

The respective nectar drops were individually streaked onto YM agar plates (1.0% glucose, 0.5% peptone, 0.3% malt extract, 0.3% yeast extract, 2.0% agar) with 0.01% chloramphenicol, and incubated at 25 °C until microbial colonies were detectable (2–20 days). A total of 158 yeast isolates was obtained from the 439 nectar drops plated. Agar plates were observed under a microscope at 10×–40× magnification (Olympus CX31) and phenotypically different yeasts were purified by streak-plating; approximately 1–5 yeast types grew per agar plate. A single clone (an entire colony) of each purified morphotype was used for species identification. As many yeast isolates from nectar drops as possible

were DNA sequenced. The large subunit (26S) ribosomal DNA gene (D1/D2 region) was two-way sequenced for each clone using the primer combination NL1–NL4, according to *Kurtzman & Robnett (1998)* and *Lachance et al. (1999)*. Raw sequences were edited and assembled and consensus sequences were obtained using Geneious Pro 8.1.7 bioinformatics software (Biomatters Ltd., Auckland, New Zealand). Nucleotide collection databases at GenBank were queried with the Basic Local Alignment Search Tool (BLAST; *Altschul et al., 1997*) to look for named yeast species with DNA sequences matching those obtained from the isolates. All sequences queried yielded significant correlations with named yeast accessions in GenBank databases, generally with 98–100% of sequence coverage and identity. Resulting DNA species and the associated sampling information (Drop, Individual, Species, Genus and Botanical family) was used for analyses of yeast diversity. The yeast isolates studied are maintained in the Centro de Investigacion Cientifica de Yucatan (CICY); their corresponding DNA sequences have been deposited in GenBank under the accessions listed in Table 1.

## Cell counts and nectar sugar composition and concentration

The density of yeast cells in each nectar drop was estimated using a Neubauer chamber and standard cell count procedures (*Herrera et al., 2009*). The initial volume of nectar drops was measured with calibrated micropipettes (*Dafni, 1992*), then each nectar sample was diluted with 0.5% lactophenol cotton blue solution to obtain a final volume of up to 1.5–6 times the initial volume. Each diluted sample was loaded on a counting chamber and examined under a microscope. Cells were counted in each of 16 quadrants of the counting chamber and cell density was calculated using the formula: cells per $\mu$L $=$ average number of cells counted in the quadrants multiplied by the dilution factor and the fixed volume of the chamber.

The sugar composition and concentration of nectar was measured using procedures described by *Herrera, Pérez & Alonso (2006)* and *Canto et al. (2011)* and ion-exchange high-performance liquid chromatography (HPLC). Samples of nectar were individually blotted onto a 10 mm $\times$ 12 mm sterile Whatman 3 MM paper wick; immediately after absorption, wicks were placed into sterile envelopes and stored at 25−26 °C in silica gel. For the analytical procedure, nectar-containing wicks were individually placed into Eppendorf tubes and 1 mL of HPLC-grade water was added to each tube. Each diluted sample was filtered using a 0.4 $\mu$m polyvinylidene difluoride (PVDF) filter and 5 $\mu$L of solution injected into a Dionex DX 500 HPLC system (Dionex; Sunnyvale, CA, USA). The HPLC system was equipped with an effluent degas module, a GP 40 gradient pump, a CarboPac PA10 (4 mm $\times$ 50 mm) guard column and a CarboPac PA10 (4 mm $\times$ 250 mm) analytical column. It also had an ED40 electrochemical detector for pulsed amperometric detection in integrated amperometric mode, with the normal preloaded wave form for sugar detection (Dionex Corp., 1994). The column was eluted isocratically (flow rate 1 mL min$^{-1}$) with 40 mM NaOH (50% solution; JT Baker, Deventer, Netherlands) and kept at 24 °C during analysis. The concentrations of sucrose, glucose and fructose in each nectar sample (g of solute per 100 mL solution) were calculated by integrating the area under the corresponding chromatogram peaks, using linear regression models fitted to the data of

standard sugar solutions, then calculating the expected concentration values corresponding with the integrated area of each sugar type in the analyzed samples. Two independent HPLC measurements were performed on each diluted sample; replicate results were averaged for the analyses.

## Data analysis

To characterize the species diversity of nectar yeasts and to compare diversity estimates across the hierarchical sampling levels (i.e., Drop, Individual, Species, Genus, Family), the analytical framework suggested by *Chao et al. (2014)* was implemented using the R package iNEXT (*Hsieh, Ma & Chao, 2016*). This method generalizes the sample size-based approach of *Colwell et al. (2012)* and the coverage-based approach of *Chao & Jost (2012)* to produce and expand rarefaction-extrapolation curves of species based on Hill numbers (*Hill, 1973*). Hill numbers are a mathematically unified family of diversity indices, differing among themselves only by an exponent $q$. These indices provide a suitable framework for measuring diversity because (1) they are expressed in units of effective numbers of species, (2) by using algebraic transformation, they are easily associated with key diversity indices such as Shannon entropy and Gini-Simpson index, and (3) their estimations can be effectively generalized to incorporate hierarchical levels of diversity in a species assemblage (*Chao et al., 2014*). For each sampling level (Drop, Individual, Species, Genus, and Family), an incidence matrix was built by recording the presence or absence across sampling units of each of the 158 DNA species identified. The first three Hill numbers (*Hill, 1973*), which are associated with estimators of species richness and species dominance, were calculated for each level; their corresponding rarefaction and extrapolation curves were constructed. The first Hill number ($q = 0$) used in the analysis estimates the expected yeast species richness (number of species) in the assemblage of nectar host plants. The second Hill number ($q = 1$) is the exponential of the Shannon entropy index and estimates yeast diversity with respect to equally common species and species richness (Shannon diversity). The third Hill number ($q = 2$) is the inverse Simpson concentration index and measures the dominance of yeast species in the species assemblage (Simpson diversity); see *Hill (1973)* for further details about Hill numbers. To compare hierarchical sampling levels, rarefaction and sample size-based extrapolation were produced for each level to provide asymptotic estimators of diversity based on Hill numbers with their respective 95% confidence intervals constructed by a bootstrap method (*Chao et al., 2014*). One potential issue in our sampling is that it included many different plant species, each with a relatively low replication. To account for this as much as possible, first, all yeast species that occurred only once were excluded from the analysis, as they were likely to be allochthonous; second, an analysis of sampling completeness was conducted to estimate the sample size needed for the proportion of undetected autochthonous species to remain unchanged even when the sample size increases (*Chao & Jost, 2012*). To this end, a sample completeness curve was constructed by combining the sample size-based and the coverage-based estimations. Extrapolations were extended up to double the initial sample size (i.e., 122 nectar samples) for all sampling levels, which allowed us to make predictions about the yeast diversity that can be detected in each sampling level using a similar sampling effort. The number of

nectar samples examined in each level was 122, 54, 17, 13, and 10 for Drop, Individual, Species, Genus, and Family, respectively.

Correspondence analysis was conducted using the R package ca (*Greenacre, Nenadic & Friendly, 2016*) to obtain a statistical and graphical visualization of associations between nectar-living yeasts and host plants. This analysis is a geometric technique for displaying the rows and the columns of a contingency table as points in a low-dimensional space such that the positions of the row and column points are consistent with their associations in the table. The analysis produces correspondence-dimensions based on the profiles (relative frequency of yeast taxa corresponding with the respective host plant), weighted average of profiles (centroid of the space representation), chi-square Euclidean-distances (proximity between points), and the total inertia (total contribution of yeast taxa and host plant to the between-taxa correspondence). For yeasts and host plant data, contingency tables were produced using yeast species as column variables and plant species as row variables. All singletons were excluded from the analysis. The first three dimensions obtained from the analysis were plotted to generate biplots representing correspondence between yeast and host plant taxa.

Given that the relationship between response and explanatory variables follows a power pattern (e.g., the response variable is proportional to the explanatory variable raised to a power), a power regression model was used to test the association between yeast cell density (explanatory variable) and nectar sugar concentration (response variable) in nectar samples. To construct the power model and test it, first the logarithm of both variables was taken and plotted to verify the linear pattern; then, a linear regression was performed on the transformed data to test the relationship between variables. The inverse transformation was made on both sizes of the linearized function to obtain the power function and the exponential term (*Rossiter, 2016*). Data were plotted taking the logarithms of both variables. To identify the contribution of different types of yeasts and host plants after removing the variance due to yeast cell density a least-square regression with two categorical co-factors was performed on the transformed data. Different groups of yeasts (Yeast) and different host plant species (Plant) were treated as co-factors. Sample sizes in several combinations of yeasts and host plants were less than five and yeast groups tended to not occur across all host plants, therefore, the Yeast was classified into groups to obtain a robust analysis. The groups of yeasts were *Metschnikowia*, *Papilotrema*, *Ustilago*, and Other yeasts. The *Metschnikowia* group included the closely related *Metschnikowia ipomoeae*, *M. koreensis*, *M. lochheadii*, and *Metschnikowia* sp. Similarly, the *Papilotrema* included *Cryptococcus laurentii var. laurentii*, *P. nemorosus,* and *P. rajasthanensis.* The *Ustilago* group included *Ustilago sparsa* and *Ustilago* sp. The yeast species with very small sample sizes were included in the Other yeasts group. Given that data are structured as an incomplete design, an interaction term (Yeast × Plant) was added to test multiplicative effects of yeasts and host plants, rather than additive effects. A Type III approach for unbalanced data was used to calculate the sums of squares (*Zahn, 2010*). The Akaike Information Criterion (AIC) was applied to measure the goodness of fit of the model, taking into account the number of parameters included and to find the best model that fits the data with the minimum number of parameters. The AIC analysis drops terms from the full model and compares the original model to the

reduced one. Analyses were calculated separately for sucrose, glucose, and fructose. In four cases, nectar samples produced more than one yeast species. In each of those cases, the yeast identity assigned in the analysis was selected at random from the co-occurring yeast species. Analyses were performed with R software (*R Development Core Team, 2016*).

## RESULTS

### Yeast diversity

A total of 39 species of yeasts was identified, composed of 48% Ascomycota and 52% Basidiomycota (Table 1). The number of colonies produced by each nectar drop is reported in the raw data file and the number of nectar drops by host plant species is reported in the Appendix S1. There was a single yeast species per nectar drop in practically all cases; two or three different yeast species occurred in only four nectar samples (see Data S1). The most frequent ascomycetous yeasts were *Metschnikowia koreensis* ($n = 13$), *M. lochheadii* ($n = 11$), and *Kurtzmaniella cleridarum* ($n = 12$), and the most frequent basidiomycetous yeasts were *Ustilago* species ($n = 14$) *Cryptococcus laurentii* var. *laurentii* ($n = 12$), and *Sympodiomycopsis paphiopedili* ($n = 8$). Analysis of diversity predicts that the overall species richness of yeasts in the sampled nectar community (Hill number $q = 0$) was between 25 and 34 species, which was in the same order of magnitude as the number of equally common species ($q = 1$, 22–34 species) or dominant species ($q = 2$, 19–33 species). Rarefaction and extrapolation curves were consistent in showing that several yeasts remained unrecorded at the Genus and Family sampling levels of the plant community surveyed. None of the three diversity estimates used reached an asymptote at those levels of the sampling hierarchy. At the Species level, species richness reached an asymptote at a sample size doubling the initial sampling effort, i.e., $n = 17$. Analyses also showed that the number of species harbored at the Drop and Plant levels was nearly completely sampled since the three estimators of diversity reached an asymptote at approximately 100 and 50 sampling units, respectively. The maximum predicted species values were 25 for species richness ($q = 0$), 22 for equally common species ($q = 1$) and 19 for dominant species ($q = 2$). At all levels, estimation of the species richness is roughly comparable to the dominance.

Rarefaction and extrapolation curves also allow us to make two predictions of Hill numbers for equally common species ($q = 1$) and dominance ($q = 2$) of yeasts in the host plant community. In the first scenario, Drop and Individual sampling categories for nectar collection reach an asymptote and harbor relatively low yeast diversity. In the second, Species, Genus and Family categories do not reach an asymptote; even when extrapolations double initial sample size and remain relatively high, there is unrecorded yeast diversity. These last categories have the highest predicted diversity of yeasts (Fig. 1, $q = 0$, $q = 1$, $q = 2$). Completeness curves show that sample completeness was nearly achieved with the current sample size at the Drop and Individual levels (1 and 0.99, respectively). At the Species level, sampled completeness was close to one (0.89) and at higher-order levels, the maximum sample completeness was 0.76 and 0.65 for Genus and Family, respectively (Fig. 1, sampling completeness).

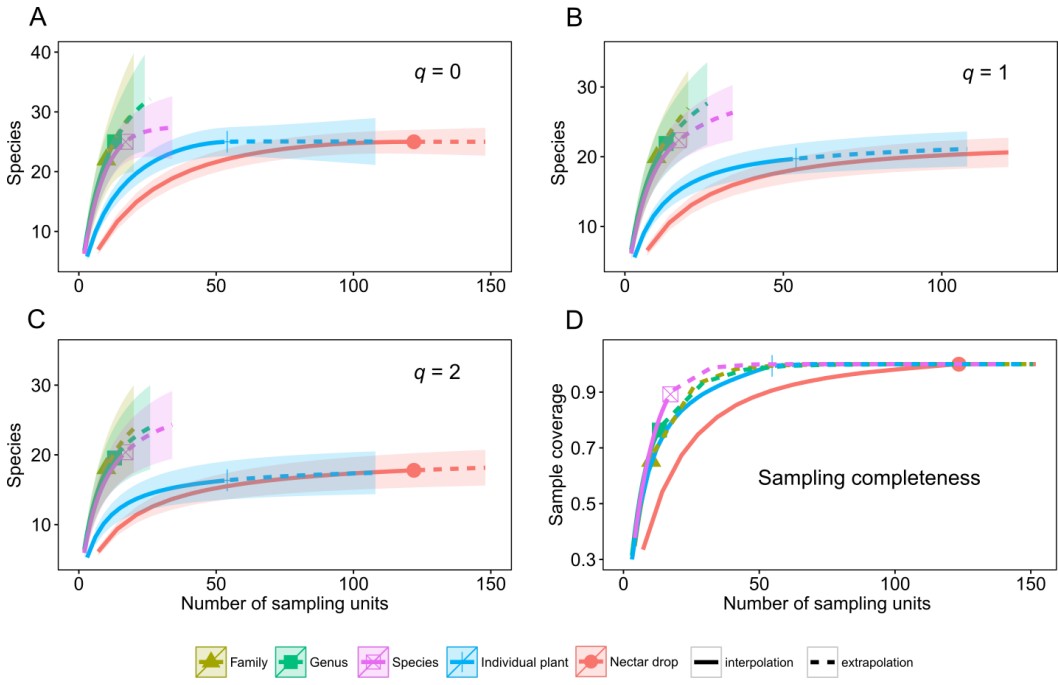

**Figure 1** **Diversity of tropical nectar-living yeasts.** Diversity of nectar-living yeasts at the different hierarchically nested sampling levels used in nectar collection: nectar drops (Drop), individual plants (Individual), plant species (Species), plant genus (Genus), and botanical family (Family). Plots show (A) species richness (Hill number for $q = 0$), (B) equally abundant species ($q = 1$), (C) dominance ($q = 2$), and (D) sample completeness curve. Diversity curves were constructed using rarefied (solid lines) and extrapolated nectar samples (dashed lines) with sample size-based estimations. Each curve was extrapolated up to double the initial sample size. Observed sample size for each category curve is denoted by a different symbol. The 95% confidence intervals (color-shaded regions) were obtained by a bootstrap method based on 200 replications.

## Yeast-plant associations

Correspondence analysis revealed a significant number of associations between yeasts and host plants (Fig. 2). The most extreme correspondence was observed between *K. cleridarum* with *Opuntia dillenii*, followed by *Starmerella* sp. and *Metschnikowia ipomoeae* with the host plant *Ipomoea crinicalyx*, *Clavispora lusitaniae* with *Agave angustifolia*, and *M. koreensis* with *Tecoma stans*. Looser associations included *Candida sorbosivorans* with *Passiflora foetida*, *Metschnikowia* sp. with *Lonchocarpus longistylus*, *Sporidiobolus ruineniae* with *Merremia dissecta*, *Papilotrema flavescens* with *Bravaisia berlandieriana*, and *Kwoniella mangrovensis* with *Operculina pinnatifida*. The weakest associations were observed between *Saitozyma flava*, *Ustilago sparsa*, and *Ustilago* sp. with *Ipomoea hederifolia* and *Ipomoea triloba*, and *Vishniacozyma taibaiensis* and *Naganishia liquefaciens* with *Piscidia piscipula* (Fig. 2).

## Yeast effects on nectar sugars

Nectar samples containing yeasts had lower average concentrations of sucrose, glucose, and fructose than nectar samples lacking yeasts, irrespective of the yeast species and host plant (Table 2). In general, significant relationships were found between yeast cell density

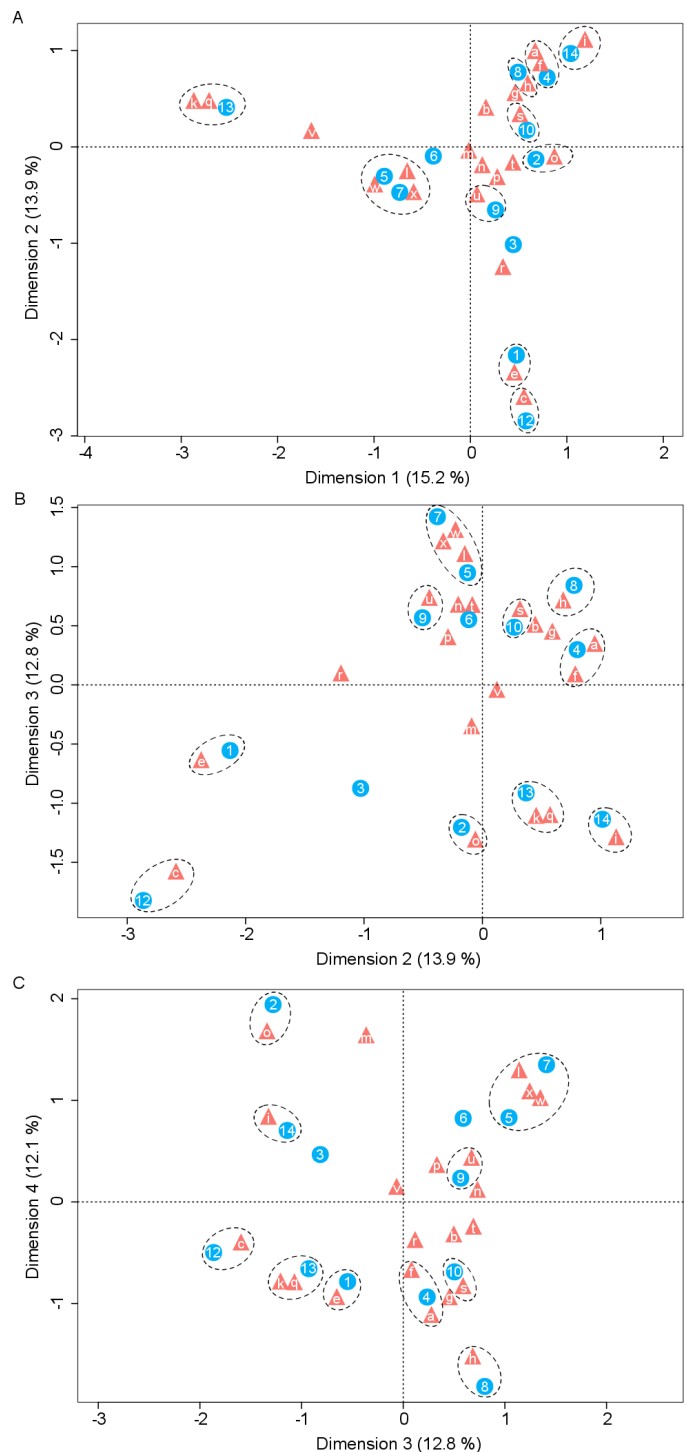

**Figure 2  Correspondence analysis for nectar-living yeasts.** Correspondence analysis for yeasts isolated from floral nectar samples. Plots show the associations between yeasts and host plants: (A) dimensions 1 and 2, (B) dimensions 2 and 3, (C) dimensions 3 and 4. Yeasts are depicted by filled triangles and letters and host plants by filled circles and numbers. Distances among points depict the similarity between members of the same yeast group or of the same plant group. 

**Figure 2 (…continued)**
White dashed-line ellipses indicate significant ($p < 0.05$) correspondences between yeasts and host plants. Percentage of contribution of each dimension to total variation is shown in parenthesis in the respective dimension. The points depicting the extreme correspondence of *Kurtzmaniella cleridarum* with *Opuntia dillenii* were extracted from the graphic analysis so that the correspondences are better observed. Yeasts: (a) *Starmerella* sp., (b) *Wickerhamiella occidentalis*, (c) *Candida sorbosivorans*, (e) *Clavispora lusitaniae*, (f) *Metschnikowia ipomoeae*, (g) *Metschnikowia lochheadii* (h) *Metschnikowia* sp., (i) *Metschnikowia koreensis*, (k) *Vishniacozyma taibaiensis*, (l) *Saitozyma flava* (m) *Cryptococcus* sp.1, (n) *Cryptococcus laurentii* var. *laurentii*, (o) *Papiotrema flavescens*, (p) *Cryptococcus* sp.2, (q) *Naganishia liquefaciens*, (r) *Hannaella siamensis*, (s) *Kwoniella mangrovensis*, (t) *Rhodotorula paludigena*, (u) *Sporidiobolus ruineniae*, (v) *Sympodiomycopsis paphiopedili*, (w) *Ustilago sparsa*, (x) *Ustilago* sp. Host plants: (1) *Agave angustifolia*, (2) *Bravaisia berlandieriana*, (3) *Gymnopodium floribundum*, (4) *Ipomoea crinicalyx*, (5) *Ipomoea hederifolia*, (6) *Ipomoea nil*, (7) *Ipomoea triloba*, (8) *Lonchocarpus longistylus*, (9) *Merremia dissecta*, (10) *Operculina pinnatifida*, (12) *Passiflora foetida*, (13) *Piscidia piscipula*, (14) *Tecoma stans*.

**Table 2  Nectar samples with yeasts and nectar samples without yeasts.** Comparisons of average ($\pm$SD) concentrations of the three nectar sugars between nectar samples containing yeast cells and nectar samples without cells. $t$-tests and statistical significance are shown for each sugar.

| Nectar sugars | Nectar samples (g of solute per 100 mL solution) | | $t$ | $d.f$ | $p$ |
|---|---|---|---|---|---|
| | **With yeasts** | **Without yeasts** | | | |
| Sucrose | $6.3 \pm 7.8$ | $10.6 \pm 11.1$ | 4.48 | 215 | <0.0001 |
| Glucose | $2.5 \pm 3.2$ | $5.5 \pm 3.8$ | 7.95 | 196 | <0.0001 |
| Fructose | $2.9 \pm 2.9$ | $5.6 \pm 3.5$ | 8.81 | 172 | <0.0001 |

and nectar sugar concentration. The decrease in concentration of sucrose, glucose and fructose were proportional to the increase in yeast cell density raised to a power coefficient. In Fig. 3, data are plotted taking the logarithms of both variables to show the linearized pattern and the power function fitted for each sugar. Different yeasts groups (Yeast) and different plant species (Plant) as main factors showed no contributions to explaining variance in the model, but the interaction between both terms had a significant impact on the relationship between yeast cell density and nectar sugar concentration (Table 3). The AIC values confirmed that the multiplicative impact of the interaction between Yeast and Plant was more important to the regression model than the additive effect of each factor. The best power model that fits the data is one that includes yeast cell density as a predictor of nectar sugar concentration and a multiplicative effect of the interaction between yeasts and host plant species (Table 3). To illustrate the interaction between Yeast and Plant factors and its impact on nectar sugar concentration, along with the overlap effect of yeast cell density, scatter plots for representative yeast species and their respective host plant species are shown in Fig. 4.

## DISCUSSION

No other studies of nectar-living yeasts have been conducted in tropical nectariferous plants to date, excepting *Herrera et al. (2009)* and *Canto & Herrera (2012)*, where the frequency of yeasts in floral nectar samples was assessed in three regions, two in southern Spain and one in southern Mexico. However, the diversity of nectar yeasts was not explicitly addressed in these previous studies, although their results suggest differences between temperate and

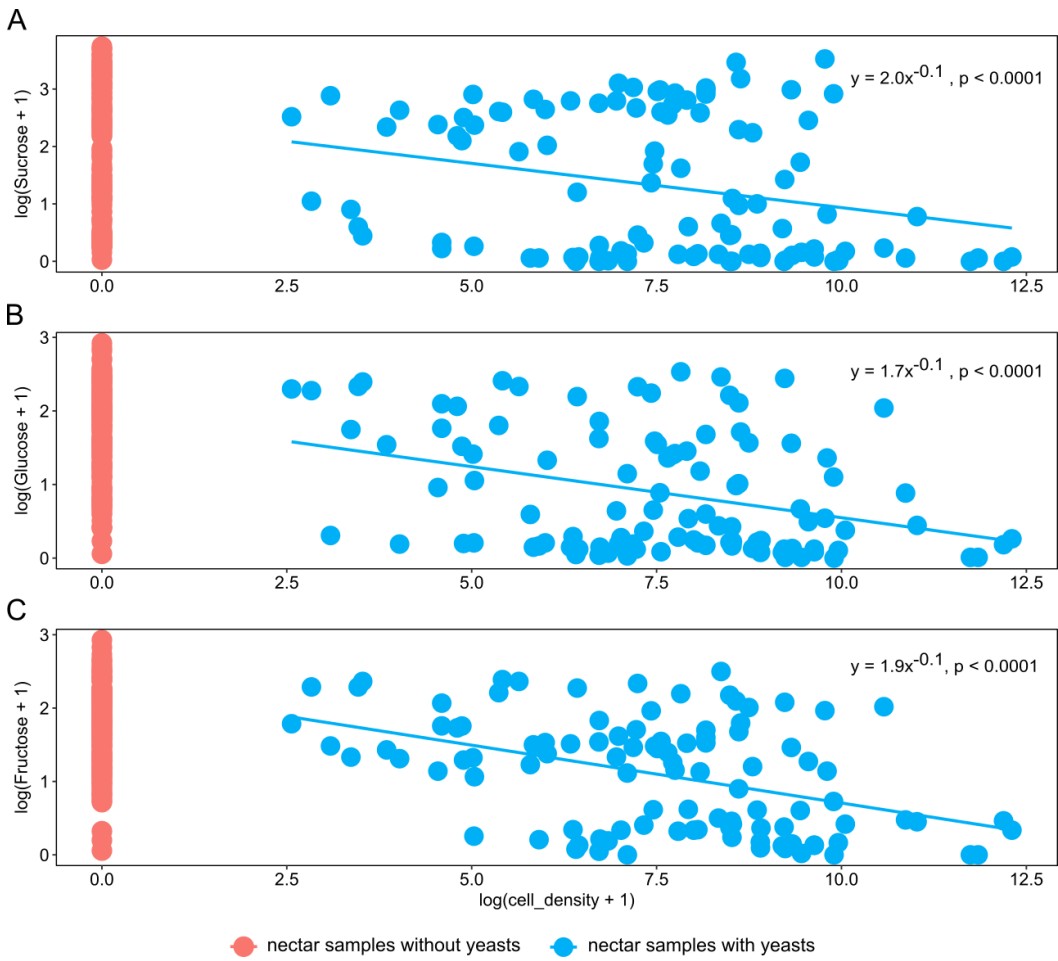

**Figure 3** **The relationship between nectar sugars and yeast cell density.** Overall relationship between yeast cell density and the concentration of nectar sugars: (A) sucrose, (B) glucose and (C) fructose. Power models are shown in each panel along with their statistical significance. The percentages of variance of each sugar explained by each model (adjusted $R^2$) are 10%, 30%, and 36% for sucrose, glucose, and fructose, respectively.

tropical regions. A similar study was conducted by *Mittelbach et al. (2015)* in a subtropical environment of the Canary Islands. We will first discuss diversity patterns found in the present study and then compare them with previous findings. Finally, we will discuss the association between yeast species and host plants and the implications of differential yeast effects on nectar sugars.

## Yeast diversity

Our results indicate that the assemblage of yeasts in the plant community surveyed was composed of a relatively high number of species at the highest sampling levels (plant genera and botanical families), along with a substantial number of equally common species and relatively low species dominance. This tropical plant community harbored a higher diversity of nectar yeasts than our sampling design could detect. While the expected yeast diversity at the drop and individual levels was estimated acceptably with the sample size set
**Table 3  Type III least-square analyses and Akaike Information Criterion (AIC) values for the effect of yeast cell density on the concentration of nectar sugars.** Sum of squares (SS), degrees of freedom ($d.f.$), $F$-values ($F$) and statistical significance ($P$-value) for the co-factors Yeast (different groups of yeasts) and Plant (different host plant species) are shown jointly with their respective AIC value. The lower the AIC value, the better the model fits to the data when a variable/co-factor is included in the model.

| Model terms | Sucrose | | | | | Glucose | | | | | Fructose | | | | |
|---|---|---|---|---|---|---|---|---|---|---|---|---|---|---|---|
| | SS | $d.f$ | F | P-value | AIC | SS | $d.f$ | F | P-value | AIC | SS | $d.f$ | F | P-value | AIC |
| Full model | | | | | −126 | | | | | −206 | | | | | −190 |
| Yeast cell density | 1.35 | 1 | 5.88 | 0.0178 | −120 | 0.84 | 1 | 7.85 | 0.0064 | −197 | 0.56 | 1 | 4.48 | 0.0375 | −186 |
| Yeast | 0 | 0 | | | −126 | 0 | 0 | | | −206 | 0 | 0 | | | −190 |
| Plant | 0 | 0 | | | −126 | 0 | 0 | | | −206 | 0 | 0 | | | −190 |
| Yeast × Plant | 4.74 | 8 | 2.57 | 0.0155 | −117 | 2.39 | 8 | 2.81 | 0.0089 | −194 | 2.62 | 8 | 2.64 | 0.0133 | −180 |
| Residual | 0.48 | 73 | | | | 0.33 | 73 | | | | 0.35 | 73 | | | |

in this study, the analysis predicts that diversity increased remarkably at higher levels in the sampling hierarchy. Reducing the number of nectar drop replicates per plant, as well as the number of individual plants per species, while increasing the number of plant genera and families will probably achieve a more encompassing picture of diversity of nectar-living yeasts in tropical plants.

A frequent pattern of animal and plant diversity is the latitudinal gradient of species richness (*Pianka, 1966*; *Hillebrand, 2004*). Although latitudinal clines in species richness are discernible in several groups of marine bacterioplankton and phytoplankton microorganisms (e.g., *Fuhrman et al., 2008*; *Schattenhofer et al., 2009*; *Barton et al., 2010*), microbial diversity has been less studied in these clines, particularly for diversity associated with tropical floral nectars. Although more studies are necessary, our results and those of the other studies reveal a possible tendency for lower latitudes to support more nectar-living yeast species than higher latitudes. For example, *Herzberg, Fischer & Titze (2002)* studied microfungal diversity in the nectars of native plants in temperate communities of Germany, reporting a species richness of 20 yeasts in a total of 25 different plant species. *Pozo, Herrera & Bazaga (2011)* found 12 yeast taxa in 24 plant species in southern Spain; later, *Álvarez Pérez & Herrera (2013)* found 20 yeasts in nectar of 30 plant species in a large plant assemblage from southern Spain. Most recently, *Mittelbach et al. (2015)* reported nectar fungal diversity from a subtropical plant community in the Canary Islands. A total of 34 yeasts species were found in eight native plant species. *Belisle et al. (2014)* reported 38 microfungi species, associated with mouthparts of 21 hummingbirds and six bat species of Costa Rica. In this work in a tropical environment, 18 nectariferous plants were surveyed and a total of 39 yeast taxa were found. Therefore, yeast species richness seems to steadily decrease from the tropical community of Yucatan and subtropical community in the Canary Islands to the temperate plant communities of southern Spain and Germany.

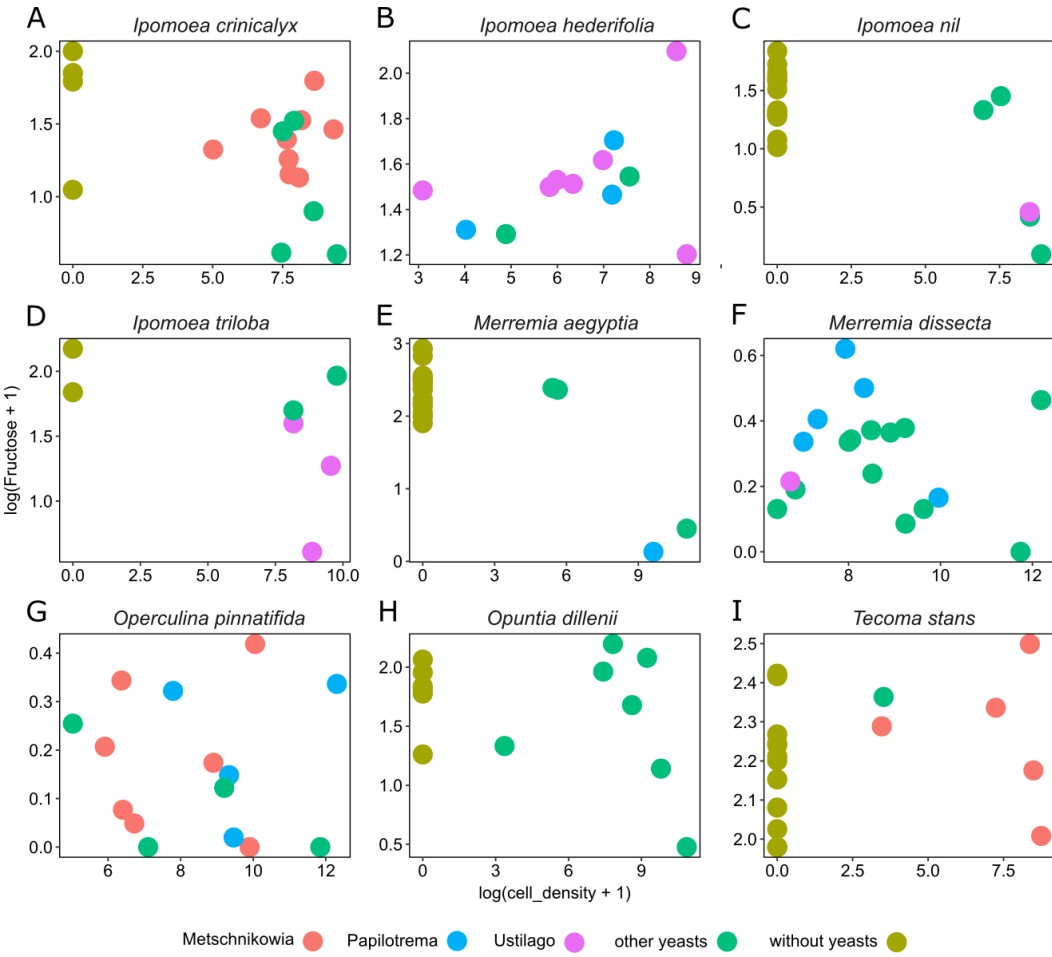

**Figure 4** **Differential effects of nectar-living yeasts.** The effect of the interaction of different types of yeasts and host plants on the relationship between yeast cell density and the concentration of sugar in nectar samples. Data are shown only for fructose concentration in nectar of (A) *Ipomoea crinicalyx*, (B) *Ipomoea hederifolia*, (C) *Ipomoea nil*, (D) *Ipomoea triloba*, (E) *Merremia aegyptia*, (F) *Merremia dissecta*, (G) *Operculina pinnatifida*, (H) *Opuntia dillenii*, and (I) *Tecoma stans*. Not all yeasts are distributed in all host plants; some yeasts seem to be specific to some plants.

## Yeast-plant associations

The diversity of nectar-living yeasts in our sample was also shaped by associations between yeast, host plants and flower visitors. This pattern creates a mosaic of nectar environments at the community level where habitat features are filters that influence the probability that the taxa, with their specified traits, can join and persist as members of a local community (*Soininen, 2012*; *Hillebrand & Blenckner, 2002*). According to our results and previous evidence (e.g., *Lachance et al., 2001*; *Lachance & Starmer, 2008*; *Lachance, Hurtado & Hsiang, 2016*), two types of non-exclusive filters may influence nectar-yeast interactions. First, floral nectar may act as a yeast community filter because of its physicochemical and nutritional factors such as availability of nutrients, water activity and the presence of yeast limiting/enhancing solutes, which can together lead to physiological specialization in nectar-living yeasts (*Lievens et al., 2015*). Our results show the existence of frequent

yeast and host plant correspondences, which is compatible with the existence of nectar filters that 'sieve' yeasts arriving to nectar and drive yeast distribution across host plants. However, experimental evidence culturing yeasts under different nectar environments are necessary to test the existence of this type of filter. Second, flower visitors can also be seen as an ecological filter as they show particular associations with yeasts. Different plant species have different pollinators that can transport different yeast species to floral nectars. In a preliminary nectar-yeast assessment in South African plants, *de Vega, Herrera & Johnson (2009)* observed that differences among plant species in yeast incidence were related to variations in pollinator types. *Mittelbach et al. (2015)* also found that differences in pollinator types partly explained variation in nectar yeast composition between Canary Islands plants. Pollinators of plants sampled for this study included solitary bees, stingless bees, hummingbirds, beetles, and bats. Thus, it seems reasonable to postulate that these different groups will carry different yeast species, and the closest yeast-plant correspondences are also caused by particular flower visitors carrying particular yeasts to flowers. For example, correspondence between *K. cleridarum* and the cactus *O. dilleni* is explained by the association of this yeast with beetles of the genus *Carpophilus*, which contact cactus flowers to feed on nectar and pollen and release yeast cells to this environment (*Lachance & Starmer, 2008*). Correspondence of *Starmerella* sp. and *M. ipomoeae* with *I. crinicalyx* denote that the flower visitors are bees and nitidulid beetles (*Rosa et al., 2003*; *Lachance et al., 2001*). The association of *M. ipomoeae* and *M. lochheadii* with *Ipomoea* species results from the association of these yeasts with *Conotelus* beetles (*Lachance et al., 2001*). In contrast, looser yeast-plant correspondences involved mostly basidiomicetous yeasts (except *C. sorbosivorans*) isolated in non-flower, non-nectar substrates and probably arrive to nectar through accidental contamination or air dispersal (*Lachance et al., 2001*; *Valério, Gadanho & Sampaio, 2002*; *Fell & Tallman, 1980*; *Yang et al., 2010*). Additionally, plant-yeast species correspondences mostly involved ascomycetous yeasts. In fact, ascomycetous yeasts showing correspondence with plants all belong to the same Saccharomycetes class (subphylum Saccharomycotina), while basidiomycetous taxa isolated from nectar belong to several classes such as Tremellomycetes, Ustilaginomycetes, Microbotryomycetes, and Hyphomycetes (subphyla Agaricomycotina, Pucciniomycotina, and Ustilaginomycotina).

## Yeast effects on nectar

Our results show that the overall effect of yeast cell density on nectar sugars generally involves changes in the composition of nectar sugars that denote not only a chemical signature of yeast metabolism but also a nectar quality impoverishment since the sugar concentration decreases with increasing yeast cell density. This phenomenon has been reported previously by *Herrera, García & Pérez (2008)* and *de Vega & Herrera (2013)*. By reducing the nutritional value of nectar, the foraging behavior of pollinators is affected and nectar-living yeasts become a factor that drives plant–pollinator interactions (*Herrera, Pozo & Medrano, 2013*; see also *Vannette, Gauthier & Fukami, 2013*; *Good et al., 2014*; *Schaeffer & Irwin, 2014*). Although more data from additional tropical communities are needed, it is reasonable to expect that nectar-living yeasts will have ecologically significant

implications in plant–pollinator interactions at the community level because of their effects on community-wide floral nectar traits and the foraging behavior of flower visitors. The results from this study also show that nectar alteration by yeasts is not a rare phenomenon in the community of host plants and is probably more frequent in tropical plant communities than is currently acknowledged.

Yeast cell density and the interaction between different yeast groups and host plants account for most of the variance observed in nectar sugar concentration in this study. Although different yeast groups were not found to have different impacts on nectar traits, their interaction with host plants impacted nectar sugar concentration. One explanation is that the initial sugar concentration of nectar depends on the variance inherent to plant species in their nectar secretion. Nectar-living yeasts can match or mismatch with traits of initial nectar (e.g., because of physiologic requirements of yeasts), therefore, different types of yeasts will differ in their ability to grow in different nectars (*Herrera, Pozo & Bazaga, 2014*). Moreover, floral nectars frequently contain plant metabolites that prevent yeast degradation of nectar (*Adler, 2000*; *Thornburg et al., 2003*; *Herrera et al., 2010*; *Heil, 2011*; *Nepi et al., 2012*). The result is that some types (or species) of yeast will occur in specific host plants but will not occur in others. This pattern was observed across host plants in this study. For example, *Metschnikowia* group yeasts occur in *I. crinicalyx*, *O. pinnatifida* and *T. stant* but did not occur in the rest of the host plants. Similarly, *Papilotrema* group yeasts occurred only in *I. hederifolia*, *M. aegyptia*, *M. dissecta* and *O. pinnatifida*, and *Ustilago* group yeasts occurred only in *I. hederifolia*, *I. nil*, *I. triloba*, and *M. dissecta*.

The observed diversity of nectar-living yeasts in the assemblage of host plants surveyed most likely represent only a portion of the actual number of species occurring in floral nectar in the area, suggesting that tropical communities harbor an impressive, as yet undiscovered diversity of yeast taxa associated with flower-nectar environments. The diversity of these types of yeasts is not only characterized by an important number of equally common species with low dominance but also by significant species correspondences between yeasts and nectariferous plants. Finally, the impact that the interaction between different types of yeasts and nectariferous plants exert on nectar sugars observed in this study suggests the existence of a nectar filtering process that sieves the initial assemblage of yeast species arriving to nectar from pollinators mouthparts, thus creating the opportunity for yeast ecological specialization.

## ACKNOWLEDGEMENTS

We thank Pilar Bazaga and Esmeralda López for assistance with DNA sequencing of yeasts; Atzelby López, Blanca Lizama, Cesar Canché and Raymundo González for assistance in the field and laboratory; Marina García for chemical analyses; Paulino Simá, Filogonio May and Alfredo Dorantes for host-plant identification.

### Funding

This work was supported by the Consejo Nacional de Ciencia y Tecnología through CB-2007-01 program (grant number 80031), the Ministerio de Educación y Ciencia (grant number CGL2010-15964) and the Junta de Andalucía (grant number P09-RNM-4517). The funders had no role in study design, data collection and analysis, decision to publish, or preparation of the manuscript.

### Grant Disclosures

The following grant information was disclosed by the authors:
Consejo Nacional de Ciencia y Tecnología: 80031.
Ministerio de Educación y Ciencia: CGL2010-15964.
Junta de Andalucía: P09-RNM-4517.

### Competing Interests

The authors declare there are no competing interests.

### Author Contributions

- Azucena Canto conceived and designed the experiments, performed the experiments, analyzed the data, contributed reagents/materials/analysis tools, wrote the paper, prepared figures and/or tables, reviewed drafts of the paper.
- Carlos M. Herrera conceived and designed the experiments, analyzed the data, contributed reagents/materials/analysis tools, wrote the paper, prepared figures and/or tables, reviewed drafts of the paper.
- Rosalina Rodriguez performed the experiments, analyzed the data, contributed reagents/materials/analysis tools, reviewed drafts of the paper.

### Field Study Permissions

The following information was supplied relating to field study approvals (i.e., approving body and any reference numbers):

Permission to collect from natural areas of the Yucatan was granted by Secretaría del Medio Ambiente y Recursos Naturales, Delegación Yucatán-Subsecretaría de Gestion para la Protección Ambiental: Dirección General de Vida Silvestre (oficio 00837/09).

### DNA Deposition

The following information was supplied regarding the deposition of DNA sequences:

The group large-subunit (26S) ribosomal DNA gene (D1/D2 region) sequences used here are provided in a Supplemental File.

### Data Availability

The raw data has been supplied as a Supplementary File.

## Supplemental Information

Supplemental information for this article can be found online at http://dx.doi.org/10.7717/peerj.3517#supplemental-information.

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
