# Peer review of "Nectar-living yeasts of a tropical host plant community: diversity and effects on community-wide floral nectar traits"

_PeerJ, doi:10.7717/peerj.3517_

## Round 0.1 · original submission · Major Revisions

The reviewers and I agree that your manuscript reports on an important and interesting topic. However, there is disparity between the reviewers about how well your manuscript accomplishes its goals. My own opinion is a compromise between those of the reviewers. While I think you have an opportunity to greatly improve the manuscript by addressing the comments of the reviewers, I am sympathetic to the critique of reviewer 1 that the methodology should be much more transparent so that readers can truly understand how much merit to place in your results. I also share the skepticism of reviewer 1 about whether the regression on nectar sugar concentration is truly meaningful. The statistical results do not match the figures, and the very small number of data points makes the analysis suspect. Please address this issue very clearly. You will either need to convince skeptical reviewers about the validity of that analysis, or use an alternative analysis that lacks the problems of the regression analysis. For example, you might try classifying sugar concentrations as high or low categories and analyzing this binomial variable. In addition to these large issues, please address each of the comments of the reviewers and check the language as closely as you can for grammatical mistakes, of which there are many.

Reviewer 1 ·

Basic reporting

The following details are missing from the methods section:
1. What is a nectar drop? (Line 100) How are three obtained from each flower? Were they pooled, then split or each obtained and treated separately?
2. Samples were extracted in 1 mL of water, but only 5 uL of that was filtered and injected? How was this small of a sample filtered? Perhaps a larger volume was filtered but only 5 ul was injected?
3. How many samples per species were used? Please include as a table
4. How can yeast species be included in the model on sugar concentrations? Were there ever multiple yeast species isolated from the same sample? This would be expected in at least a few samples. How did the authors deal with this type of sample in the regression analysis?
5. How many samples/species plated? How many colonies sequenced per species? How many colonies sequenced per morphotype? Was there a 1:1 correlation between consensus ID and morphotype?

Experimental design

The study is a survey. Sampling details are not fully described, as outlined above. For example, how were flowers chosen? Were all samples taken at one site or across multiple sites? Further, it is unclear if the plant species were sampled at the same time of the year, or peak flowering etc. or if they are comparable.

Validity of the findings

Data are taken from a previous paper, and additional analyses are performed. Although some of the analyses appear to have been done properly (but see comments on Fig 3), these additional analyses are not fully justified for a few reasons outlined below.

1) In the current study, diversity patterns are reported at the level of nectar drop, individual plant, species and genus. However, it is unclear how many samples were examined at each level, and if sampling was equal across time, space, flowering age etc. among plants. The number of samples per plant is reported in Canto and Herrera 2012, but is not repeated here (even in the supplement). Even if the number of samples taken per plant or flower were reported, we would also need to know how many colonies per sample were sequenced to interpret results. Finally, rarefaction is reported at the level of the nectar drop. However, individual drops were not re-sampled. Instead, samples from all plant species were used as replicates. Interpreting what this analysis means biologically is unclear. Is this the diversity that is estimated to occur at the level of the individual nectar drop averaged across plant species?
2) The authors seem to assume that there is only a single yeast species per flower/nectar drop, but do not present any evidence for this claim. Further, the sampling method used--sequencing from isolated colonies-- would only be able to examine this if many or most of the colonies on the plate were sequenced (it is unclear if they are) or if cell morphotype was also used (and validated) as a method to determine species per sample.
3) The assumption above (single yeast species per nectar drop) was used in the analyses, including the regressions were performed linking yeast density within a sample based on the yeast species IDed from that sample. As stated before, no justification for this assumption is given. In addition, this regression also assumes that the species found in the nectar is the one that is responsible for any metabolic breakdown of sugars. However, pollinators vector many microbial taxa, including yeasts and bacteria, and extinction is also possible in this system, so the assumption although likely, is not tested that the identified species is also responsible for sugar metabolism.
4) Figure 3 reports the p-values for a regression between yeast density and sugar concentration in the nectar sample. The p-values included here are somewhat suspect. For example, what looks like a positive correlation (Figure 3a) is not significant, while a panel with 4 points that do not form any sort of a line has a p-value of 0.031. The statistics on this figure are should be checked. In addition, a regression should not even be run with only 4 points, as is done in three of the panels. Finally, there are two fit lines but only one statistic reported for Cryptococcus laurentii. Is the fit line for both species together or one of the fit lines or the model as a whole? This entire analysis is questionable for reasons outlined in 2-4 above.
5) Are the authors confident that isolated yeasts rather than other microorganisms or variation in floral traits etc. are responsible for change in nectar sugars? What about organisms that may not be culturable given the current methods?

Additional comments

The paper is generally well-written and increased study of tropical nectar yeast communities is definitely welcome. This paper overlaps substantially with a previously published paper by these authors (Canto and Herrera 2012) and uses most of the same data. This would be acceptable if the analyses were justified. However, the new analyses in this paper are not valid given how the data were collected and rely on assumptions that the authors do not address as outlined above (single yeast species per nectar sample and that yeast communities are fully sampled). It is unclear if enough work has been done to merit an additional publication from these data.

Reviewer 2 ·

Basic reporting

Overall, the manuscript is mostly clear and well-structured. Below, I highlight instances where points could be made more clear:

Lns 54-57: While the host nectar environment can certainly impact what species are capable of colonizing, your sentence suggests that microbial colonization order can also be important in determining community composition, as well as variation in nectar features. Perhaps worthwhile to edit this sentence to make this point more clear, and cite Peay et al. 2012 (Proc Royal) and Vannette & Fukami 2014 (Ecology Letters).

Ln 135: Change “counting” to “counted”

Ln 137: Edit sentence to state “The SUGAR composition and concentration….”

General comment regarding units; please be consistent in either including a space or not between a number and units reported throughout the manuscript. In other words, 1mL vs. 1 mL

Ln 271: Change “cells” to “cell”

Ln 272: Change “yeasts” to “yeast”

Ln 273: Change “cells” to “cell”

Lns 304-321: I think this paragraph as a whole could be cleaned up a little. Instead of jumping into a laundry list of results from other studies on findings of yeast diversity, perhaps prepare the reader with a better topic sentence. I think your statements on there being latitudinal clines in species richness could be moved to the front of this paragraph, and perhaps highlight that these clines for microbial diversity have been less studied (but see examples from marine systems). Then perhaps jump into your discussion of prior findings from temperate systems vs. your own. Just a suggestion.

Lns 370-371: To be thorough, perhaps cite Schaeffer & Irwin 2014 (Ecology) which also looked at nectar yeast impacts on plant-pollinator interactions

Check format for Jacquemyn reference (journal title)

Check format for Pianka reference (report full journal title)

Figure 1 caption: Missing period at end of last sentence. Also, please make the following edits:

“...between MEMBERS of the same yeast….”

“Whiting dashed-ellipses INDICATE significant correspondence between yeasts and host plants.”

Table 1 caption: Edit to state “...yeast species in nectar is REPORTED along with..”

Figures: In terms of overall presentation of a manuscript and results, I think it looks best if color schemes are consistent across figures. For instance, a couple figures have grey axes lines, while the other has black. Different color schemes between Fig. 1 and 2 also clash for me. Perhaps use the same blue and red pastels for Fig 2 as in 1? Just a suggestion, which should be easy to implement in ggplot2.

Experimental design

The intent of their study is well-defined, and methods employed are sound. They are clearly defined, and build off of extensive experience and prior work on related questioning in other systems. I also commend them on their use of Hill numbers and Chao's framework for addressing diversity in this study.

Validity of the findings

Lns 312-313: Please double-check your reporting on yeast species richness and how many plant species were sampled. In the Methods/Results you state that 39 yeast species from 24 plant species were identified. In the Discussion here however, these numbers do not match up (41 yeast species from 18 plant species).

---

## Round 0.2 · Major Revisions

This new version of the manuscript is much improved, although it still needs some substantial work to make it ready for publication. I agree with the new comments and suggestions from reviewer 1, and I ask that you address each of these as thoroughly as possible. (I did not send it back to reviewer 2, who was generally positive the first time around.) In order to avoid another round of minor revisions after this one, please do try to have a native English speaker read the final draft of your revised manuscript.

I look forward to receiving the second revision! It is an interesting piece of work!

Reviewer 1 ·

Basic reporting

Language is generally very good and to a high standard. The literature cited is thorough and appropriate. Data are shared and the paper is organized appropriately.

I suggest that the authors have an colleague who speaks English as a first language also read through the article, as there are still many grammatical issues throughout, especially in the newly added sections (e.g. lines 45-48, lines 221-233 etc etc).

Experimental design

The study is meaningful and interesting to the broader audience. The technical details of sampling are much better explained now and are appropriate for publication.

The new statistical analyses still require improvement before publication. Specifically, Figure 1 and the accompanying analysis or the rationale and interpretation still require improvement. Specifically, the goal of this analysis and its interpretation are unclear and I think perhaps incorrect. The current analysis suggest that sampling done at the drop level suggest that the community should have ~25 species and was nearly complete. By differentially pooling these data into matrices summarized by plant species (supplementary information), the authors suggest that increased sampling of plant species or genera would dramatically increase the number of yeast species in the community. However, this logic seems not quite correct and not justified in the manuscript. Instead, it is my understanding that the authors’ goal was to compare diversity among plant species, which would require a different input matrix to iNEXT. If within-plant species diversity was of interest (which I think that it is), the input files would be either drop-level or plant-level measures of yeast diversity from a single plant species which would be used for calculating sampling curves for each species. Individual curves from each species could be compared by examining overlap of confidence intervals and could address if different plant species host more or less diverse communities. But if I am incorrectly interpreting this figure and its goal is to highlight that communities are diverse, I think that the different sampling levels are unnecessarily confusing and generate misleading results due to sample pooling.

In addition, More details are necessary to fully describe the regression presented in Table 3. What is a power regression and why was it used? It is my understanding that a power regression had an exponential term in the regression? The reference cited (Zahn 2010) does not mention power regression in the paper. Was the ‘total AIC ‘ the AIC of the model with no predictors included? For the model with interaction term, were previous factors (Yeast, Plant, Cell density etc) also included in the model? I believe that they should be but the df do not change as a result of changing predictors so perhaps the authors need to check the values in the table?

Validity of the findings

See comments above on statistical analyses used. Clarification would be very helpful.

In addition, there is one thing that raised a question in my mind:
nectar samples had very high cell densities (supplementary table 1), so it seems surprising that the authors only recovered between 1-5 colonies per plate (lines X_ , particularly if the total nectar drop was plated. Do the authors have any hypotheses to explain this? Also, other cell counts indicated extremely high cell densities in some samples but no colonies were recovered. Perhaps it the case that only a few (1-5) morphotypes were described per sample, but the total number of colony forming units was much higher?

The authors suggest that there is a yeast x plant species interaction in effects on sugars, but could there be an interaction between yeast density and plant species, where yeasts differentially influence nectar sugars in some plant species and not others? This seems like a more appropriate analysis, since not all yeast species were observed in each plant species.

Additional comments

Thank you for your hard work revising this manuscript. The manuscript is much improved and the addition of Figure 4 is welcome. These data are extremely valuable and should be shared with a broader audience. I encourage the authors to aim to improve presentation and rationale for the analyses, particularly Fig 1 and accompanying discussion (lines 327-329 and surrounding) and the new regression analysis as noted above.

---

## Round 0.3 · accepted · Accept

I commend you for your excellent work in revising the manuscript in terms of content and presentation quality. I made a small number of final edits to the text of the manuscript to correct grammatical mistakes, and I have attached this version to the decision (link is below). The most notable change is in the title, in which I changed "tropical plant host community" to "tropical host plant community" to be consistent with the rest of the manuscript.

---

## Author Rebuttal · Round 0.3

June 05th, 2017

Professor Michael Singer
Academic Editor, *PeerJ*

Dear Editor,

We appreciate the time and efforts of the reviewers in this second review, as well as their thoughtful comments on the manuscript. As detailed below, we have addressed all concerns and suggestions pointed out by the Reviewer 1 and yourself. We hope you find this version acceptable for publication in *PeerJ*.

Sincerely yours,

[Figure]

Azucena Canto
Researcher of the Centro de Investigacion Cientifica de Yucatan (CICY)

On behalf of all authors
* * *
### *Responses to reviews*

To help you follow changes made to the manuscript, our responses are given as sentences in blue below the passages corresponding to comments and suggestions (italicized text here) made by the reviewers. Line numbers refer to the revised version.

**Editor's Comments**

MAJOR REVISIONS

*This new version of the manuscript is much improved, although it still needs some substantial work to make it ready for publication. I agree with the new comments and suggestions from reviewer 1, and I ask that you address each of these as thoroughly as possible. (I did not send it back to reviewer 2, who was generally positive the first time around.) In order to avoid another round of minor revisions after this one, please do try to have a native English speaker read the final draft of your revised manuscript.*

As detailed below, we have addressed all concerns raised, and incorporated changes suggested to the manuscript. The language has been reviewed by the Elsevier Language Editing Services (see Elsevier letter below).
* * *
***Reviewer 1 (Anonymous)***

*I suggest that the authors have an colleague who speaks English as a first language also read through the article, as there are still many grammatical issues throughout, especially in the newly added sections (e.g. lines 45-48, lines 221-233 etc etc).*

Done. The manuscript has been edited by Elsevier Language Editing Services (see Elsevier letter below).

*Experimental design*

*The new statistical analyses still require improvement before publication. Specifically, Figure 1 and the accompanying analysis or the rationale and interpretation still require improvement. Specifically, the goal of this analysis and its interpretation are unclear and I think perhaps incorrect. The current analysis suggests that sampling done at the drop level suggest that the community should have ~25 species and was nearly complete. By differentially pooling these data into matrices summarized by plant species (supplementary information), the authors suggest that increased sampling of plant species or genera would dramatically increase the number of yeast species in the community. However, this logic seems not quite correct and not justified in the manuscript. Instead, it is my understanding that the authors' goal was to compare diversity among plant species, which would require a different input matrix to iNEXT. If within-plant species diversity was of interest (which I think that it is), the input files would be either drop level or plant-level measures of yeast diversity from a single plant species which would be used for calculating sampling curves for each species. Individual curves from each species could be compared by examining overlap of confidence intervals and could address if different plant species host more or less diverse communities. But if I am incorrectly interpreting this figure and its goal is to highlight that communities are diverse, I think that the different sampling levels are unnecessarily confusing and generate misleading results due to sample pooling.*

We are agreeing in that the rationale and interpretation of the analysis accompanying Figure 1 were not concordant with the goal stated in the text and that is why results, Fig. 1 and the respective discussion were confusing. The goal of our diversity analysis was to know how diverse the community of nectar-living yeasts is in a tropical host plant community following a hierarchical sampling design and comparing between different levels in that hierarchy. We have thoroughly revised and edited all those parts of the text related with this analysis to give a concordant context for the diversity analysis conducted. We think that our analysis of diversity is correct, the problem rather being that it was not concordant with the expressed objective. The specific question for the analysis has been thoroughly edited to clarify this point. We have also verified that results and their interpretation were concordant with the goal of the analysis. Changes have been incorporated in Abstract, Introduction, Results and Discussion sections (Lns 5-7, 14-17, 21-22, 63-65, 336-346).

*In addition, more details are necessary to fully describe the regression presented in Table 3. What is a power regression and why was it used?*

We use a power regression model because the relationship between response and explanatory variables follows a power pattern (e.g., the response variable is proportional to the explanatory variable raised to a power). A comprehensive explanation of what is a power regression model and why it was used to data analysis was added in the Materials and Methods section (Lines 225-236, 251-252). Table 3 was improved and now it shows a more complete information of the Type III least-square analyses and Akaike Information Criterion (AIC) values, degrees of freedom, which help to a better understanding of the models testing, the statistical significance of main factors and interaction term (see Table 3, title and values).

*It is my understanding that a power regression had an exponential term in the regression?*

It is correct; a power regression has an exponential term. In the Figure 3, power regression equations have been supplied with their respective exponential terms for regression relationship (yeast cell density versus sugar concentration).

*The reference cited (Zahn 2010) does not mention power regression in the paper.*

It was a misplaced reference; Zahn 2010 is the reference on which our Type III square sums are based. This reference has been now correctly placed (Line 247-248)

*Was the 'total AIC ' the AIC of the model with no predictors included? For the model with interaction term, were previous factors (Yeast, Plant, Cell density etc) also included in the model? I believe that they should be but the df do not change as a result of changing predictors so perhaps the authors need to check the values in the table?*

The "total AIC" corresponded to the AIC estimate when the model is saturated (i.e., all terms are included). The AIC analysis drops arguments from the full model one at a time and successively compares the original model to the reduced one. For the comparison between full model and model without interaction term, all previous factors were included. We have re-edited the Table 3 to clarify the changes in degrees of freedom and AIC values. We have also included brief information in the text about the way that AIC analysis works (Lns 251-252).

*Validity of the findings*

*In addition, there is one thing that raised a question in my mind: nectar samples had very high cell densities (supplementary table 1), so it seems surprising that the authors only recovered between 1-5 colonies per plate (lines X_ , particularly if the total nectar drop was plated. Do the authors have any hypotheses to explain this? Also, other cell counts indicated extremely high cell densities in some samples but no colonies were recovered. Perhaps it the case that only a few (1-5) morphotypes were described per sample, but the total number of colony forming units was much higher?.*

In some agar plates, a continuous mass of microbial growth was found, being difficult to count colonies; in other cases, very few colonies grew, therefore, for each agar plate we isolate the different morphotypes. We have clarified this important point in the Materials and Methods section (Lns 119-124).

*The authors suggest that there is a yeast x plant species interaction in effects on sugars, but could there be an interaction between yeast density and plant species, where yeasts differentially*

*influence nectar sugars in some plant species and not others? This seems like a more appropriate analysis, since not all yeast species were observed in each plant species*

The goal of this analysis was to address the main effect of yeast cell density on sugar concentration, taking into account that there are yeast species that are more frequent in some host plant species than in others. That is why we used the Type III square sums to examine the significance of each partial influence of yeasts on nectar sugars, that is, the significance of yeast cell density with all the other effects in the model. Given that if a significant interaction is present, the main effects should not be further analyzed, keeping the yeast cell density as main effect in the model.

*Comments for the Author*

*Thank you for your hard work revising this manuscript. The manuscript is much improved and the addition of Figure 4 is welcome. These data are extremely valuable and should be shared with a broader audience. I encourage the authors to aim to improve presentation and rationale for the analyses, particularly Fig 1 and accompanying discussion (lines 327-329 and surrounding) and the new regression analysis as noted above.*

Thank you for your comments. We have addressed as thoroughly as possible all suggestions and comments. We have edited the discussion to improve the analysis for yeast diversity. We have changed the suggested lines and surrounding to become a more concordant discussion (Lns 336-346).
* * *
[Figure]

**Language Editing Services**

*Registered Office:*
Elsevier Ltd
The Boulevard, Langford Lane,
Kidlington, OX5 1GB, UK.
Registration No. 331566771

# To whom it may concern

The paper "Nectar-living yeasts of a tropical host-plant community: diversity and effects on community-wide floral nectar traits" by Azucena Canto was edited by Elsevier Language Editing Services.

Kind regards,

Biji Mathilakath
**Elsevier Webshop Support**

(This is a computer generated advice and does not require any signature)